# Autonomic Function in Obese Children and Adolescents: Systematic Review and Meta-Analysis

**DOI:** 10.3390/jcm13071854

**Published:** 2024-03-23

**Authors:** Georgios E. Papadopoulos, Foteini Balomenou, Xenofon M. Sakellariou, Christos Tassopoulos, Dimitrios N. Nikas, Vasileios Giapros, Theofilos M. Kolettis

**Affiliations:** 1Department of Cardiology, Faculty of Medicine, University of Ioannina, 45110 Ioannina, Greece; georgios.e.papadopoulos@gmail.com (G.E.P.); xensakel@gmail.com (X.M.S.); xtass@ymail.com (C.T.); dimitrios.nikas@gmail.com (D.N.N.); 2Department of Child Health, Faculty of Medicine, University of Ioannina, 45110 Ioannina, Greece; f.balomenou@uoi.gr (F.B.); vgiapros@uoi.gr (V.G.); 3Cardiovascular Research Institute, 45110 Ioannina, Greece

**Keywords:** obesity, heart rate variability, autonomic nervous system, sympathetic activity, vagal activity, children, adolescents

## Abstract

**Background:** Obesity is invariably accompanied by autonomic dysfunction, although data in pediatric populations are conflicting. **Methods:** We conducted a systematic review and meta-analysis of 12 studies (totaling 1102 participants) comparing obese and normal-weight subjects (5–18 years of age), defined as body mass index >95th or <85th percentile, respectively. Using a random-effects model, we report the standardized mean differences (SMD) of sympathetic and vagal indices of heart rate variability. **Results:** Autonomic dysfunction was present in the obesity group, based on the average SMD in the standard deviation of sinus intervals (at −0.5340), and on the ratio of low (LF)- to high (HF)-frequency spectra (at 0.5735). There was no difference in sympathetic activity, but the heterogeneity among the relevant studies weakens this result. SMD in HF (at 0.5876), in the root mean square of successive differences between intervals (at −0.6333), and in the number of times successive intervals exceeded 50 ms divided by the total number of intervals (at −0.5867) indicated lower vagal activity in the obesity group. **Conclusions:** Autonomic dysfunction is present in obese children and adolescents, attributed to lower vagal activity. Further studies are needed in various pediatric cohorts, placing emphasis on sympathetic activity.

## 1. Introduction

Obesity is an important health-related problem worldwide. It is closely related to established risk factors of atherosclerosis, such as dyslipidemia, hypertension, and diabetes, and plays a key role in its development [1]. Atherosclerosis, associated with high morbidity and mortality rates, affects medium-sized and large arteries, leading to coronary, carotid, and peripheral vascular disease. Driven by the high worldwide prevalence of obesity, ongoing research aims at providing further insights into the underlying pathophysiology.

Autonomic dysfunction has been long-identified in obese subjects [2], and consistently reiterated in subsequent studies examining various cohorts [3]. Moreover, a recent meta-analysis of studies examining weight changes revealed higher sympathetic and lower vagal activity accompanying weight gain, which were reversed after weight loss [4]. These observations carry pathophysiologic and therapeutic implications, as alterations in both autonomic arms have been linked with the progression of atherosclerosis in adult populations [5,6]. However, it remains unclear whether this process is active during the early stages of atheromatous plaque formation. To this end, incidental findings in autopsy studies have provided evidence of atherosclerosis (mainly in the form of fatty streaks and fibrous plaques) in the aorta, carotid, and coronary arteries as early as in the second decade of life [7].

Early-life obesity has emerged as a substantial public health concern, with 30% of the global child and adolescent population classified as overweight [8]. More importantly, childhood obesity often continues during adulthood, implying a continuum in the pathogenesis of atherosclerosis [9,10]. Like in adult populations, autonomic dysfunction has been reported in obese children and adolescents, but the relative contribution of each autonomic arm is uncertain, due to diverse results in hitherto published reports [11,12]. Therefore, the aim of the present work was to further investigate the precise changes underlying autonomic function in obese pediatric populations. For this purpose, we conducted a systematic review and meta-analysis of studies comparing sympathetic and vagal indices between obese and normal-weight children and adolescents.

## 2. Materials and Methods

The methodology followed in the present work adhered to the Preferred Reporting Items for Systematic Reviews and Meta-Analyses (PRISMA) guidelines [13]; the completed PRISMA statement is provided as Appendix A. The study protocol was registered in the international prospective register of systematic reviews (PROSPERO, National Institute for Health and Care Research, York, UK) with registration number CRD42023432583.

### 2.1. Search Terms

We searched the PubMed (National Institute of Health, Bethesda, MD, USA) and Scopus (Elsevier BV, Amsterdam, The Netherlands) databases for relevant studies published in English throughout December 2023. No specific criteria were set regarding the geographic origin of the studies. The search terms included were ‘“cardiac autonomic dysfunction” OR “autonomic nervous system dysfunction” OR “autonomic dysfunction” OR “heart rate variability” OR “sympathetic nervous system” OR “parasympathetic nervous system” OR “sympathetic activity” OR “vagal activity”; AND “obesity” OR “overweight” OR “excess weight” OR “adiposity” OR “body mass index” OR “BMI”; AND “children” OR “pediatric” OR “child” OR “adolescents” OR “teenage” OR “youth”’.

### 2.2. Search Strategy and Inclusion Criteria

We screened observational, cross-sectional studies and registries of children and adolescents, 5–18 years of age, reporting autonomic indices derived from heart rate variability (HRV) analysis of electrocardiographic Holter recordings. Further to animal studies, we excluded systematic reviews, meta-analyses, case reports, and editorials. After examining the full text of pertinent articles, their references were checked for additional suitable studies. Those comparing sympathetic and vagal indices, as well as their balance, between normal-weight and obese subjects were considered eligible for inclusion in our meta-analysis. For the purposes of the present work, normal weight was defined as a body mass index (BMI) below the 85th percentile of the distribution, whereas obesity was defined as a BMI above the 95th percentile, in accordance with present guides [14].

### 2.3. Autonomic Variables

Data extraction and evaluation were conducted independently by two authors (G.E.P. and F.B.), with ensuing differences adjudicated after discussions among all. In addition to autonomic data, we extracted information on study design and clinical characteristics. Based on the current consensus [15], we extracted variables depicting autonomic balance, as well as sympathetic and vagal activity, separately. For a more comprehensive evaluation, we used an array of indices, partly overcoming the disadvantages of each individual HRV variable [15]. Thus, for the description of sympatho-vagal balance we selected (a) the standard deviation of the inter-beat interval between normal sinus beats (SDNN) after time-domain HRV analysis, with lower values indicating sympathetic dominance, and (b) the ratio of low (LF, 0.04–0.15 Hz)- to high (HF, 0.15–0.4 Hz)-frequency spectra after frequency-domain analysis of HRV, with lower values indicating vagal dominance. The peak LF and HF spectra were selected for the description of sympathetic and vagal activity, respectively. As additional variables describing vagal activity, we extracted the root mean square of successive differences (RMSSD) between normal heartbeats, and the number of times successive intervals exceeded 50 ms, divided by the total number of intervals (PNN50).

### 2.4. Evaluation of the Studies

The search results were evaluated using the previously validated Newcastle–Ottawa quality scale for observational studies [16], a collaborative project between the University of Newcastle, Australia, and the University of Ottawa, Canada. In this tool, a score is given on eight items, facilitating quick visual assessment.

### 2.5. Heterogeneity

The degree of heterogeneity was estimated using the restricted maximum-likelihood estimator, expressed as τ^2^, coupled to two additional tests, namely the Q-test and the I^2^ statistic, as previously described [17]. In cases of heterogeneity, we provide a prediction interval for the true outcomes. Studentized residuals were used to assess outliers, based on values greater than the 100(0.95/2k)^th^ percentile of a standard normal distribution (where k stands for the number of studies). Likewise, studies with a Cook’s distance greater than the median plus six times the interquartile range of the Cook’s distances were considered overly influential.

### 2.6. Standardized Mean Differences

Aided by the Jamovi software package [18], the analysis of all variables was performed using the standardized mean difference (SMD, i.e., the observed difference divided by the standard deviation) as the outcome measure, with a random-effects model fitted to the data. The corresponding SMD (with 95% confidence intervals, CIs) were calculated, followed by the construction of forest and funnel plots, whereas asymmetry of the latter was assessed using the rank correlation and regression tests. Statistical significance was defined at an alpha cutoff value of 0.05.

## 3. Results

After removal of duplicates, 814 results were identified, of which 33 were further assessed for eligibility. Finally, 12 studies [19,20,21,22,23,24,25,26,27,28,29,30] reporting values separately for each autonomic arm or their balance were included in the meta-analysis. These studies were conducted between 2001 and 2016, originating from five countries, namely Brazil (four studies), Italy (four studies), Mexico (one study), Turkey (one study), and the USA (two studies). The duration of recordings ranged from 20 min to 24 h. Table 1 shows the results of the Newcastle–Ottawa quality scale.

The PRISMA Flow Diagram used for the study selection process is shown in Figure 1.

### 3.1. Clinical Characteristics

The participants were recruited from the community in two studies [19,22], from pediatric outpatient clinics in three [26,27,29], and from both sources in two [20,30]. In one further study [28], the cohort also participated in another pediatric population study, whereas the recruitment method was not easily discernible in the remaining four studies.

The participants’ demographic and clinical characteristics are summarized in Table 2. As per protocol, BMI differed between the two populations, with values at 28.79 ± 4.18 kg/m^2^ in obese and 17.07 ± 2.36 kg/m^2^ in normal-weight subjects. The obesity group consisted of 422 subjects with a mean age of 11.6 ± 2.5 years (48.5% male), whereas the normal-weight group consisted of 680 subjects with a mean age of 9.95 ± 2.56 years (53.7% male). Obese children and adolescents had higher blood pressure readings, with respect to systolic and diastolic blood pressure. In addition, this group tended to have an unfavorable lipid profile, mainly concerning plasma triglyceride levels.

Information on physical exercise is provided by two studies [24,25], reporting no differences between groups; of note, a further study [20] reported a trend (*p* = 0.086) towards less physical exercise in obese subjects.

### 3.2. Heterogeneity

The results of all tests used for the evaluation of heterogeneity are summarized in Table 3.

For variables describing sympatho-vagal balance, an absence of significant heterogeneity in the true outcomes was present regarding the SDNN. There was no indication of outliers in the context of this model, and none of the studies could be considered overly influential. Among studies reporting LF/HF, moderate heterogeneity was present, but without major indication of outliers or overly influential reports.

There was substantial heterogeneity among studies reporting LF (describing sympathetic activity), as shown by all three relevant tests (i.e., τ^2^, Q-test, and I^2^ statistic). One study [19] had a value of studentized residuals greater than ±2.6901; thus, it may be regarded as a potential outlier and as overly influential in the context of this model.

For variables describing vagal activity, there was no heterogeneity in the true outcomes regarding HF and the PNN50, without any indication of outliers. However, one study [19] could be considered overly influential in the context of the model reporting both the PNN50 and the RMSSD. With respect to the latter variable, there was moderate heterogeneity among the relevant studies, with the regression test indicating funnel plot asymmetry. By contrast, the rank correlation and regression tests showed an absence of funnel plot asymmetry in HF and the PNN50.

The funnel plots of all variables are shown in Figure 2.

### 3.3. Autonomic (Sympatho-Vagal) Balance

In total, k = 12 studies (with a population of 1102 subjects) were included in the analysis of the SDNN and LF/HF. The observed standardized mean differences of the SDNN ranged from −0.9684 to −0.1284, with an estimated average SMD of −0.5340 (95% CI: −0.6996 to −0.3684), which differed from zero (z = −6.3219, *p* < 0.0001). The 95% prediction interval for the true outcome is between −0.8352 and −0.2328, which is in the same direction as the estimated average outcome. The forest plot of the SDNN is shown in Figure 3.

The observed SMD of LF/HF ranged from 0.0239 to 1.27, with an estimated average SMD of 0.5735 (95% CI: 0.3622 to 0.7848), which differed from zero (z = 5.3186, *p* < 0.0001). The 95% prediction interval for the true outcomes is between 0.0524 and 1.0946. Despite considerable heterogeneity, the true outcomes of the studies were generally in the same direction as the estimated average outcome. The forest plot of LF/HF is shown in Figure 4.

### 3.4. Sympathetic Activity

In total, k = 7 studies (with a population of 346 subjects) were included in the analysis of LF. The observed SMD ranged from −1.1107 to 0.7393, with 86% of estimates pointing in one direction. The estimated average SMD was 0.2684 (95% CI: −0.1973 to 0.7340), which did not differ significantly from zero (z = 1.12, *p* = 0.25). Furthermore, the 95% prediction interval for the true outcome was found between −0.8945 and 1.4313, not excluding the possibility of a true outcome in the opposite direction as the estimated average SMD. The forest plot of LF is shown in Figure 5.

### 3.5. Vagal Activity

In total, k = 7 studies (with a population of 346 subjects) were included in the analysis of HF. The observed SMD ranged from −1.1164 to −0.3264, with an estimated average SMD of −0.5876 (95% CI: −0.8146 to −0.3605), which differed from zero (z = −5.0712, *p* < 0.0001). The forest plot of HF is shown in Figure 6.

In total, k = 11 studies (with a population of 1030 subjects) were included in the analysis of the RMSSD. The observed SMD ranged from −1.3992 to −0.1450, with an estimated average SMD of −0.6333 (95% CI: −0.8502 to −0.4164), which differed from zero (z = −5.7226, *p* < 0.0001). The 95% prediction interval for the true outcome was between −1.1312 and −0.1354. Values were in the same direction as the estimated average outcome, despite the presence of moderate heterogeneity. The forest plot of the RMSSD is shown in Figure 7.

In total, k = 9 studies (with a population of 493 subjects) were included in the analysis of the PNN50. The observed SMD ranged from −1.2996 to −0.0361, with an estimated average SMD of −0.5867 (95% CI: −0.8221 to −0.3514), which differed from zero (z = −4.8857, *p* < 0.0001). The 95% prediction interval for the true outcome was found between −1.0426 and −0.1309; Values were in the same direction as the estimated average outcome, despite some heterogeneity present. The forest plot of the PNN50 is shown in Figure 8.

## 4. Discussion

In addition to its effects on blood pressure, glucose level, and lipid profile, obesity is recognized as independently related to atherosclerosis, although the underlying mechanisms for this association remain under investigation. Adipose tissue can be viewed as an endocrine organ, with activated adipokines/cytokines promoting endothelial dysfunction and hyperpermeability of the vascular intima that result in the formation of atheromatous plaques [31]. Autonomic dysfunction, accompanying systemic tissue inflammation, is thought to play a central role, as previously shown in adult populations [32]. The long course of atherosclerosis development over decades of life extends the need for exploring this process in pediatric populations [7].

In the present work, we examined autonomic activity in obese children and adolescents by performing a meta-analysis of 12 published studies evaluating autonomic function in a pooled population of ~1100 subjects. Their mean age was approximately 10.5 years, with gender participation adequately balanced. We noted a small age difference between cohorts, but it was deemed unlikely to have confounded the autonomic status. Overall, the groups were formed appropriately, as reflected in the mean BMI of 28 kg/m^2^ and 17 kg/m^2^ in obese and normal-weight subjects, respectively. Like the observations in adults, obese children and adolescents had higher blood pressure readings and a rather unfavorable lipid profile, indicating coexistence of these factors in pediatric populations as well.

### 4.1. Autonomic Balance

The correlation between autonomic dysfunction and body habitus was demonstrated nearly three decades ago in a population of 600 male military veterans, between 21 and 80 years of age [33]. Our meta-analysis demonstrates that this process is also active early in life, evidenced by a shift in autonomic balance towards sympathetic prevalence in obese children and adolescents. The degree of heterogeneity among relevant studies describing the SDNN was satisfactory, with all values pointing in the same direction. This index is favored as a marker of global autonomic activity, calculated by a straightforward methodology, which enhances the comparability across various reports [15]. Additionally, the LF/HF ratio, also widely used, pointed towards the same direction. Nonetheless, a moderate degree of heterogeneity was found among relevant studies, which was solely attributed to the LF component, as discussed below.

### 4.2. Sympathetic Activity

Further to autonomic balance, we evaluated each autonomic arm separately, driven by the distinct effects attributed to their function [34]. Regarding sympathetic activity, such information would be of pathophysiologic and clinical relevance, based on recent evidence suggesting a key role of the sympathetic nervous system in enhancing the inflammatory process and, thereby, atherosclerosis [35]. The underlying mechanisms are currently under active investigation, aiming at the advent of new therapeutic options [5].

Our meta-analysis did not reveal a significant difference in sympathetic activity, as the power of the peak LF spectrum (after frequency-domain HRV analysis) was similar in obese and normal-weight children and adolescents. However, the high degree of heterogeneity observed among the relevant studies substantially weakens this result.

It is unclear to which extent the absence of difference in LF values can be attributed to inherent methodological issues, related to frequency-domain analysis, that may have contributed to the observed heterogeneity across studies [36]. More importantly, our neutral result may reflect true variation in sympathetic activity in pediatric populations, given the previously demonstrated relation with the duration of obesity [25], a confounding factor that was not addressed here. Of note, variability regarding sympathetic activity is found also in adult series, reporting higher [37], similar [38], or lower [39] sympathetic activity in obese subjects. Lastly, we did not address the amount of physical activity, which also affects autonomic function [40], a point further discussed below. Thus, our work does not permit solid inferences regarding sympathetic activity in obese pediatric subjects. More studies including larger populations are needed, perhaps examining children and adolescents separately, given their distinct characteristics. Furthermore, incorporating additional HRV methodology (e.g., detrended fluctuation analysis) may be of value in describing sympathetic indices.

### 4.3. Vagal Activity

We report lower vagal activity in obese children and adolescents compared to their normal-weight counterparts. This conclusion is based on the consistent results of our meta-analysis involving three variables, namely the RMSSD and the PNN50 (both derived from time-domain analysis), as well as the power of the peak HF spectrum. We noted the presence of some degree of heterogeneity regarding the RMSSD and the PNN50, likely attributed to normal fluctuations of these variables [41], which may have confounded recordings of shorter duration in some reports [19,20,21,22,24,28,30]. Nonetheless, the mean values of the variables describing vagal activity were in the same direction among all studies, and, overall, the funnel plot asymmetry tests were satisfactory.

Our findings extend into pediatric populations with the observations establishing a link between lower vagal activity and obesity [37]. Such association is thought to be closely intertwined with physical inactivity in childhood and adolescence, documented in earlier studies of television viewing, followed by reports of mobile and gaming devices [42]. It is generally thought that a sedentary lifestyle, with excessive screen time at these ages, is linked to obesity through exposure to food marketing and mindless eating, whilst displacing the time spent on physical activities.

Despite the limited amount of information on exercise reported here, three points deserve attention. First, a trend (although not statistically significance) towards less physical activity was found in obese subjects in one study [20]. Second, a cohort of adolescents, also included in our meta-analysis [19], displayed marked improvement in vagal activity after a 36-session resistance-training course. Lastly, another study [43] compared four groups of children, namely obese or lean, physically active or inactive (with a mean of 9.5 years of age); normal autonomic function was found in physically active participants, irrespective of body weight, strongly suggesting that exercise can increase vagal tone in obese children [43].

Further to high sympathetic activity, low vagal tone has been correlated with the severity of coronary artery disease in adult populations [44]. In this regard, the vagus nerve exerts a pathophysiologic role in plaque formation in the coronary [45] and carotid [46] arteries, modulating atherosclerosis-related inflammation. In turn, altered baro- and chemoreceptor reflexes in the diseased carotid sinus can be apparent at early stages of atherosclerosis, a timeframe during which the carotid arteries appear as predilection sites [46]. The reduced vagal activity reported here provides further support to its role in early atherosclerosis, when viewed in the context of previous ultrasound findings of abnormal intimal-to-medial thickness in the carotid arteries of obese children and adolescents [47,48,49].

### 4.4. Limitations

Studies of autonomic function in pediatric populations are characterized by marked heterogeneity that precludes solid inferences. This can be attributed to the small number of participants in each study, with a broad range of age and Tanner’s stages, despite the well-established effects of these parameters on HRV [50]. In our meta-analysis, we aimed at providing more information on the activity of each autonomic arm, utilizing. a random-effects model that ameliorates methodological differences across various reports. However, we feel that bias could not be eliminated, particularly when interpreting the results on sympathetic activity. In addition, several confounding factors, such as physical activity, or the type (central or peripheral) and duration of obesity, were not addressed here.

### 4.5. Future Directions

The relationship between body habitus at early stages of life, sympathetic activity and atherosclerosis is only beginning to emerge, fueled by recent demonstration of complex structural artery–brain circuits in atherosclerosis-diseased arteries [35]. The clinical ramifications of such observations are eagerly awaited, examining various cohorts. Further studies are also deemed necessary in the evaluation of vagal activity in pediatric populations, including overweight subjects with a BMI between the 85th and 95th percentile of the distribution. Evaluation of cohorts in both extremes of the BMI distribution would also add valuable information, based on current data indicating a U-shaped curve, characterized by poor outcome in the lowest and highest extremes [51]. The causes of such curve are still under investigation, with autonomic dysfunction appearing as a probable explanation, especially when evident during the early stages of life. Future work should examine prepubertal and pubertal cohorts separately, considering the correlation between metabolic parameters (varying across the periods of development) and HRV [52]. The role of afferent vagal neurons in controlling metabolism merits particular attention, in view of the established relation between vagal afferent signaling and loss-of-control eating, an observation with recently instituted therapeutic implications in obese adults [53].

## 5. Conclusions

Our meta-analysis demonstrates autonomic dysfunction in obese children and adolescents, which is largely attributed to reduced vagal tone. Sympathetic activity seems to vary, with further research required for clinically meaningful conclusions. Understanding the impact of autonomic dysfunction in obesity may aid in the implementation of simple and effective therapeutic strategies that will mitigate the long-term untoward health consequences.

## Figures and Tables

**Figure 1 jcm-13-01854-f001:**
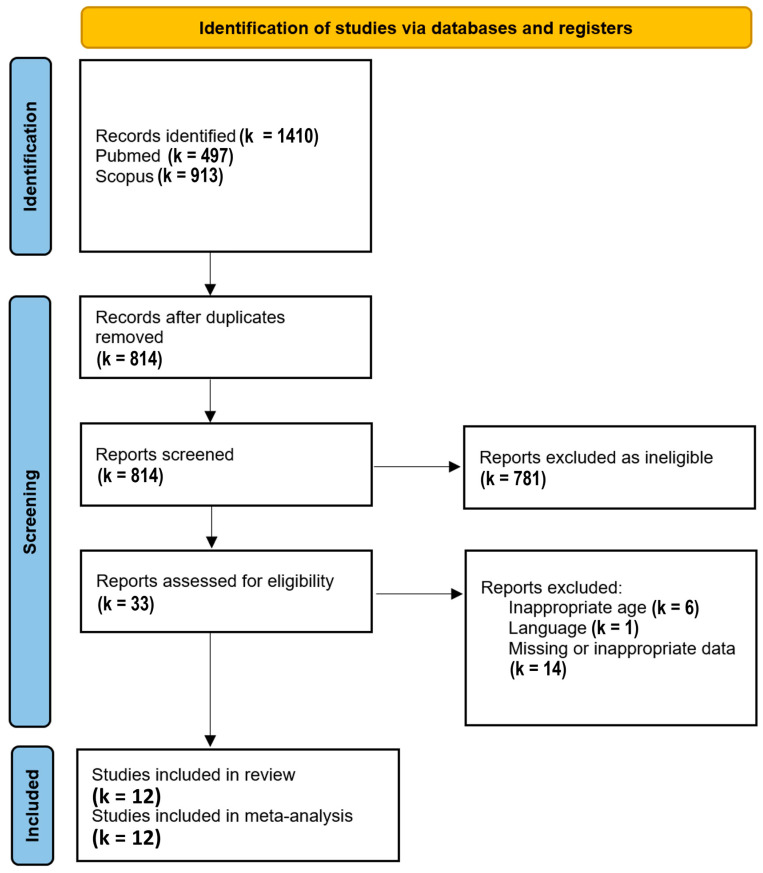
PRISMA flow diagram; k stands for the number of studies.

**Figure 2 jcm-13-01854-f002:**
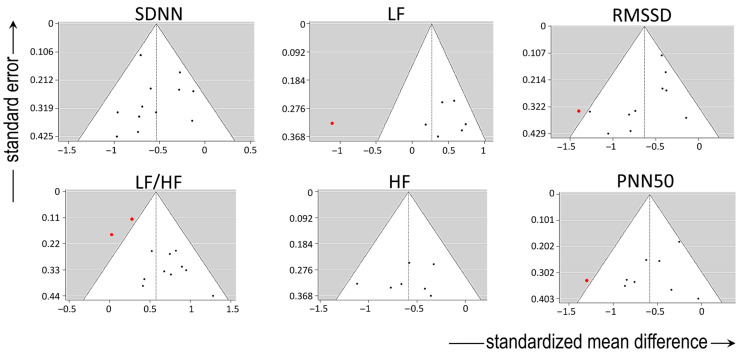
Funnel plots of all variables. Note the heterogeneity, particularly in the low-frequency (LF) spectrum and in the ratio of LF to high-frequency (HF) spectra after frequency-domain analysis. Abbreviations: SDNN: standard deviation of the inter-beat interval between sinus beats, RMSSD: root mean square of successive differences between normal intervals, PNN50: number of times successive intervals exceed 50 ms, divided by the total number of intervals. Red dots depict reports outlying the funnel.

**Figure 3 jcm-13-01854-f003:**
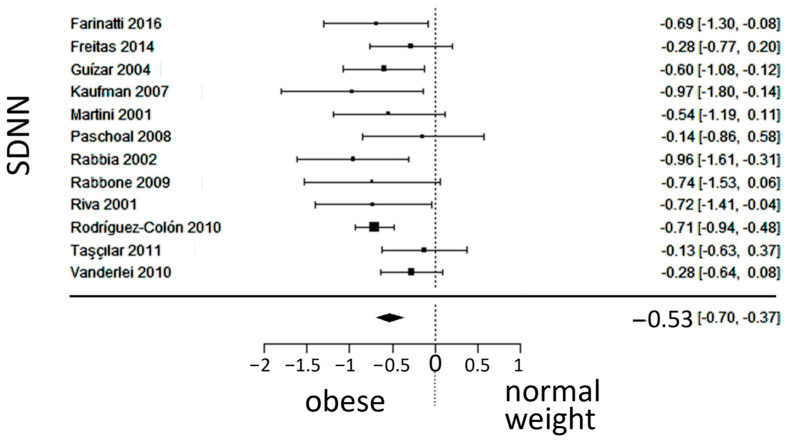
Forest plot of the standard deviation of the inter-beat interval between sinus beats (SDNN), depicting sympatho-vagal balance. The standardized mean difference (SMD) indicated the tilt of autonomic balance towards sympathetic prevalence in the obesity group. Included studies [19,20,21,22,23,24,25,26,27,28,29,30].

**Figure 4 jcm-13-01854-f004:**
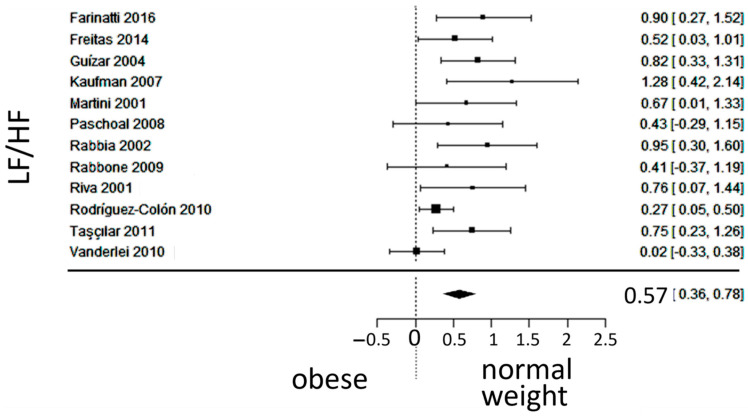
Forest plot of the ratio of low-frequency (LF) to high-frequency (HF) spectra after frequency-domain analysis. The standardized mean difference (SMD) indicated the tilt of autonomic balance towards vagal prevalence in the normal-weight group. Included studies [19,20,21,22,23,24,25,26,27,28,29,30].

**Figure 5 jcm-13-01854-f005:**
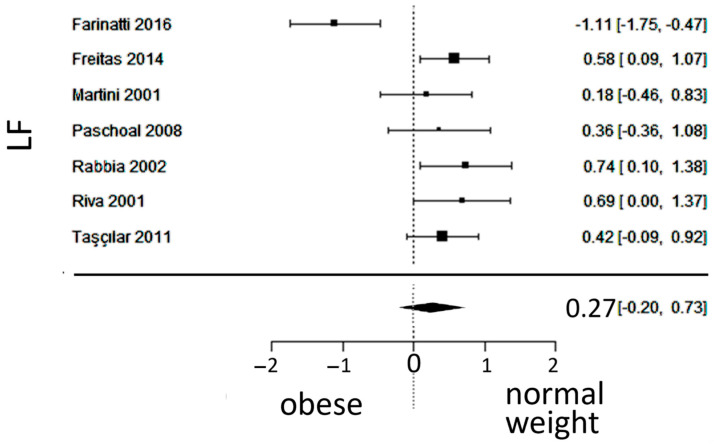
Forest plot of the low-frequency (LF) spectrum after frequency-domain analysis, depicting sympathetic activity. The standardized mean difference (SMD) failed to show a difference between the groups. Included studies [19,20,23,24,25,27,29].

**Figure 6 jcm-13-01854-f006:**
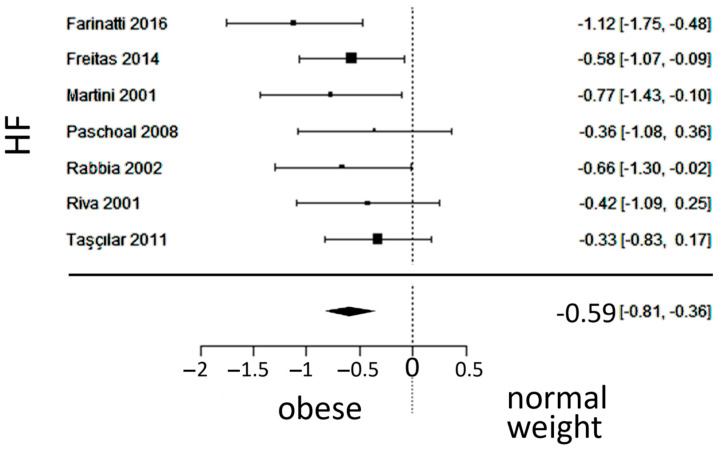
Forest plot of the high-frequency (HF) spectrum after frequency-domain analysis, depicting vagal activity. The standardized mean difference (SMD) indicated lower vagal activity in the obesity group. Included studies [19,20,23,24,25,27,29].

**Figure 7 jcm-13-01854-f007:**
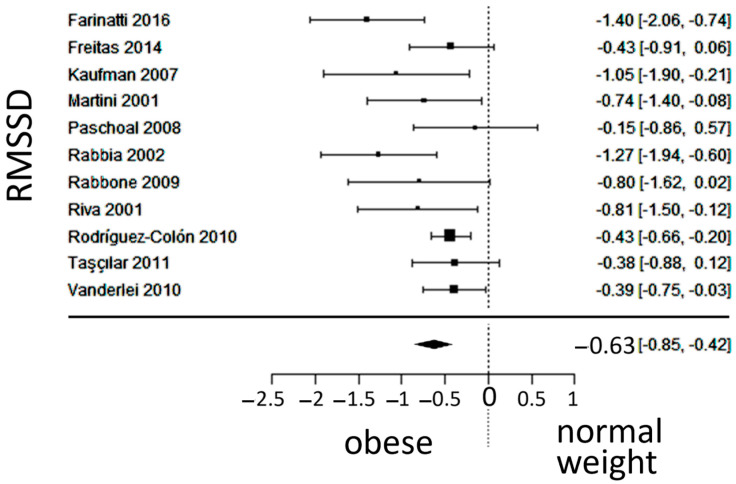
Forest plot of the root mean square of successive differences between normal heartbeats (RMSSD), depicting vagal activity. The standardized mean difference (SMD) indicated lower vagal activity in the obesity group. Included studies [19,20,22,23,24,25,26,27,28,29,30].

**Figure 8 jcm-13-01854-f008:**
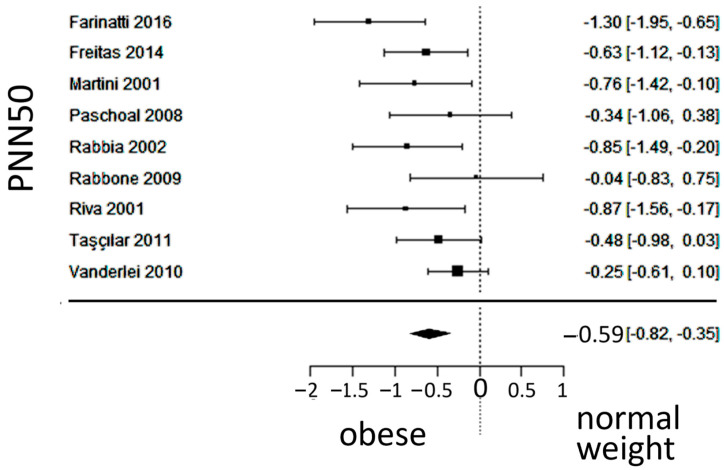
Forest plot of the number of times successive intervals exceeded 50 ms, divided by the total number of intervals (PNN50), depicting vagal activity. The standardized mean difference (SMD) indicated lower vagal activity in the obesity group. Included studies [19,20,23,24,25,26,27,29,30].

**Table 1 jcm-13-01854-t001:** Newcastle–Ottawa quality scale for the included studies.

Reference Number	[19]	[20]	[21]	[22]	[23]	[24]	[25]	[26]	[27]	[28]	[29]	[30]
Representativeness ^1^	1	1	1	1	1	1	1	1	1	1	1	1
Selection ^2^	1	1	1	1	1	1	1	1	1	1	1	1
Exposure ^3^	1	1	1	1	1	1	1	1	1	1	1	1
Baseline ^4^	1	1	1	1	1	1	1	1	1	1	1	1
Comparability ^5^	1	1	1	1	1	1	1	1	1	1	1	1
Outcome ^6^	0	0	0	0	1	0	1	1	1	0	1	0
F/U duration	N/A ^7^	N/A ^7^	N/A ^7^	N/A ^7^	N/A ^7^	N/A ^7^	N/A ^7^	N/A ^7^	N/A ^7^	N/A ^7^	N/A ^7^	N/A ^7^
F/U adequacy	N/A ^7^	N/A ^7^	N/A ^7^	N/A ^7^	N/A ^7^	N/A ^7^	N/A ^7^	N/A ^7^	N/A ^7^	N/A ^7^	N/A ^7^	N/A ^7^
**Score**	5	5	5	5	6	5	6	6	6	5	6	5

^1^ Representativeness of the exposed cohort. ^2^ Selection of the non-exposed cohort. ^3^ Ascertainment of exposure. ^4^ Baseline assessment. ^5^ Comparability of cohorts based on the design or analysis. ^6^ Assessment of outcome. ^7^ Non-applicable (N/A), as none of the studies included follow-up (F/U).

**Table 2 jcm-13-01854-t002:** Characteristics of participants. Values are given as mean ± standard deviation.

	Normal-Weight (*n* = 680)	Obese (*n* = 422)
Age, years	9.9 ± 2.5	11.6 ± 2.5
Male sex, %	53.7	48.5
BMI, kg/m^2^	17.07 ± 2.36	28.79 ± 4.18
SBP, mmHg	109 ± 11	121 ± 12
DBP, mmHg	65 ± 8	74 ± 11
TC, mg/dL	152.5 ± 26.4	163.3 ± 29.5
HDL, mg/dL	46.4 ± 11.5	47.0 ± 9.5
LDL, mg/dL	95.4 ± 19.8	91.6 ± 22.4
TGs mg/dL	82.9 ± 32.1	95.1 ± 33.4
FPG, mmol/L	84.8 ± 4.7	87.8 ± 5.6

Abbreviations: BMI: body mass index, SBP: systolic blood pressure, DBP: diastolic blood pressure, TC: total cholesterol, HDL: high-density lipoprotein, LDL: low-density lipoprotein, TGs: triglycerides, FPG: fasting plasma glucose.

**Table 3 jcm-13-01854-t003:** Heterogeneity statistics.

Variable	k	τ	τ^2^ (SE)	I^2^	H^2^	d.f.	Q	*p*
SDNN	12	0.128	0.0165 (0.0329)	20.34%	1.255	11	12.405	0.334
LF/HF	12	0.243	0.0591 (0.0569)	47.54%	1.906	11	20.272	**0.042** ^1^
LF	7	0.544	0.2956 (0.2279)	75.62%	4.102	6	23.420	**<0.001** ^1^
HF	7	0	0 (0.0532)	0%	1	6	4.649	0.590
RMSSD	11	0.229	0.0523 (0.0571)	43.79%	1.779	10	17.077	0.073
PNN50	9	0.199	0.0397 (0.063)	31.53%	1.460	8	11.767	0.162

^1^ Significant heterogeneity detected. Abbreviations: SE: standard error; d.f.: degrees of freedom; SDNN: standard deviation of the inter-beat interval between normal sinus beats; LF: low (0.04–0.15 Hz)-frequency spectrum; HF: high (0.15–0.4 Hz)-frequency spectrum; RMSSD: root mean square of successive differences between sinus intervals; PNN50: number of times successive intervals exceed 50 ms divided by the total number of intervals.

## Data Availability

Not applicable.

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
