# Peer review of "Autonomic Function in Obese Children and Adolescents: Systematic Review and Meta-Analysis"

_jcm, 2024, doi:10.3390/jcm13071854_

Round 1
Reviewer 1 Report
Comments and Suggestions for Authors
Concern exists about the level. of exercise of subjects. How were subjects selected in each country and/or study. Age of subjects is perhaps more consequential - 5-18 is great range.. these need to be delineated clearly
Reviewer 2 Report
Comments and Suggestions for Authors
The following review, “Autonomic function in obese children and adolescents: systematic review and meta-analysis,” conducted a systematic review and meta-analysis of 12 studies looking at the link between obesity and autonomic dysfunction, particularly in the pediatric population, where currently there are conflicting data sets.
It is an important topic to be focused on since obesity, particularly childhood obesity, is on the rise and has tripled since 1975, as per the World Health Organization. It is a global problem seen all across the globe. It is known that obesity is linked to autonomic dysfunction, which is thought to be one of the main contributors to hypertension, cardiac remodeling, and even death. While reviewing the paper, I stumbled upon the other systematic reviews and meta-analyses looking at the effect of weight changes on autonomic dysfunction. Having in-depth analysis and reviews of the pediatric population gives valuable insight into the connections and link between obesity and autonomic dysfunction, which in turn helps in identifying the risk factors and developing precautions and measures to take at a global public health level.
Regarding the search terms in 2.1, Where is the terminology from the area of atherosclerosis/plaques used to look at studies that involved childhood/adolescent obesity and atherosclerosis risk since the author mentions the incidence of asymptomatic plaques in the intima of the aorta, carotid, and coronary arteries as early as in the second decade of life?
Are the observational and cross-sectional studies used for this analysis specific to some geographic areas?
For this analysis, normal weight is defined as below the 85th percentile and obesity as above the 95th percentile. Did you see how the data looks if you looked at the subsets of the pediatric population who are very far below the 85th percentile for health-weighed individuals and far above the 95th percentile in the obese groups?
Since you mention that no single variable depicts autonomic conditions accurately, how did you make sure that the combined variables depicted the autonomic conditions more accurately?
I find the methods, results, and figures are concise and clearly explain each part of the analysis among all the studies used in this analysis and review.
The discussion section is clear, depicts the findings clearly, and connects them to the pathophysiology in a well-written fashion.
I applaud the fact that the authors had a limitation section where they discussed the possible limitations.
It would be great if the conclusion talked about the future directions and areas to be studied using the knowledge gained through this systematic review and meta-analysis.
Comments on the Quality of English LanguageSpell checks, spacing, and flow should be checked. Quality of English seem fine
Reviewer 3 Report
Comments and Suggestions for Authors
Dear Authors,
Autonomic function in obese children and adolescents: systematic review and meta-analysis was probably a required review. You have good results outcome of lower vagal activity rather than increased sympathetic activity. Would age contribute to not having high sympathetic activity? CHildren are playing and active.
In review terms line 68 and 69: "heart rate variability" is repeated? Any reason? Would you have to repeat search again after deleting the repeat term?
